# Predicting in-hospital mortality in ICU patients with lymphoma using machine learning models

Ling Xu[1], Guang Tu[2], Zhonglan Cai[2], Tianbi Lan [ID][3]*

**1** Breast Department, Dongguan Hospital, Guangzhou University of Traditional Chinese Medicine, Dongguan, China, **2** Department of Cardiology, Lichuan People's Hospital, Fuzhou, China, **3** Department of Hematology, The Tenth Affiliated Hospital, Southern Medical University (Dongguan People's Hospital), Dongguan, China

* tianbi_lan@126.com

## Abstract

### Background

Lymphoma is a severe condition with high mortality rates, often requiring ICU admission. Traditional risk stratification tools like SOFA and APACHE scores struggle to capture complex clinical interactions. Machine learning (ML) models offer a more accurate alternative for predicting outcomes by analyzing large datasets. However, their application in predicting in-hospital mortality for lymphoma patients remains limited.

### Objective

This study aims to develop and validate machine learning models to predict in-hospital mortality in ICU patients with lymphoma using data from the Medical Information Mart for Intensive Care IV (MIMIC-IV) database, thereby enhancing risk stratification and clinical decision-making.

### Methods

We conducted a retrospective cohort study using data from the MIMIC-IV database, which includes detailed clinical data from adult patients admitted to the ICU. Patients with a primary diagnosis of lymphoma were included. Baseline characteristics, laboratory parameters, and clinical outcomes were extracted. Lasso regression was employed to screen for significant risk factors associated with in-hospital mortality. Fifteen machine learning models, including logistic regression, random forest, gradient boosting, and neural networks, were developed and compared using receiver operating characteristic (ROC) curves and area under the curve (AUC) analysis. Model performance was evaluated through cross-validation and SHapley Additive exPlanation (SHAP) values to interpret variable importance.

**Data availability statement:** The datasets analyzed during this study are derived from the publicly available MIMIC-IV v3.1 database (https://physionet.org/content/mimiciv/3.1/). Credentialed researchers can obtain access by completing the CITI Data or Specimens Only Research course and submitting a data use agreement via PhysioNet. All data are de-identified and HIPAA-compliant, and the analysis code along with variable extraction scripts have been deposited in the public GitHub repository (https://github.com/tianbilan/MIMIC-IV-Lymphoma-Mortality-ML).

**Funding:** The author(s) received no specific funding for this work.

**Competing interests:** The authors have declared that no competing interests exist.

**Abbreviations:** AIDS, Acquired Immunodeficiency Syndrome; AUC, Area Under the Curve; BUN, Blood Urea Nitrogen; CatBoost, Categorical Boosting; DBP, Diastolic Blood Pressure; ICU, Intensive Care Unit; IQR, Interquartile Range; ML, Machine Learning; ROC, Receiver Operating Characteristic; SD, Standard Deviation; SBP, Systolic Blood Pressure; SHAP, SHapley Additive exPlanations; SpO$_2$, Oxygen Saturation; WBC, White Blood Cell Count.

## Results

A total of 1591 patients were included, with 342 (21.5%) in-hospital deaths. Lasso regression identified significant predictors of mortality, including blood urea nitrogen (BUN), platelets, PT, heart rate, systolic blood pressure, APTT, spo2, and bicarbonate. The CatBoost Classifier demonstrated the highest predictive performance with an AUC of 0.7766. SHAP analysis highlighted the critical role of BUN as the most important factor in mortality prediction, followed by platelets and PT. The SHAP force plot provided individualized risk assessments for patients, demonstrating the model's ability to identify high-risk subgroups.

## Conclusion

Machine learning models, particularly the CatBoost Classifier, effectively predict in-hospital mortality in ICU patients with lymphoma. These models outperform traditional statistical methods and provide valuable insights into risk stratification. Future work should focus on external validation and clinical implementation to improve patient outcomes in this high-risk population.

## Introduction

Lymphoma is a severe condition that often requires admission to the intensive care unit (ICU) due to its associated high mortality rate and complex clinical presentation [1]. The pathophysiology of lymphoma involves a combination of immune dysregulation, metabolic disturbances, and potential complications such as infections and organ failure [2–5]. Early identification of patients at high risk of in-hospital mortality is crucial for optimizing treatment strategies and improving outcomes. Traditional risk stratification tools, such as the Sequential Organ Failure Assessment (SOFA) score and the Acute Physiology and Chronic Health Evaluation (APACHE) score, have limitations in capturing the complex interactions among multiple clinical variables [6–8]. Machine learning (ML) models offer a promising alternative by leveraging large datasets to identify patterns and predict outcomes more accurately [9]. ML has been increasingly applied in various medical fields, including disease diagnosis, treatment optimization, and prognosis prediction [10,11]. However, its application in predicting in-hospital mortality for patients with lymphoma remains limited. This study aims to develop and validate ML models to predict in-hospital mortality in ICU patients with lymphoma using data from the Medical Information Mart for Intensive Care IV (MIMIC-IV) database, thereby enhancing risk stratification and clinical decision-making.

Compared to the general ICU population, lymphoma patients present distinct mortality determinants that traditional risk scores (e.g., SOFA, APACHE) fail to capture. Rapid tumor lysis syndrome (TLS) precipitates acute kidney injury and electrolyte derangements [12], while profound chemotherapy-induced immunosuppression predisposes to opportunistic infections such as Pneumocystis jirovecii pneumonia and invasive aspergillosis [13]. Direct organ infiltration by lymphoma (e.g., renal or

hepatic parenchyma) and cardiotoxicity from anthracycline-based regimens further complicate clinical trajectories [14]. These lymphoma-specific pathophysiological features create complex, non-linear interactions among variables, rendering generic ML models inadequate. Yet, ML tailored specifically to lymphoma ICU patients remains unexplored, highlighting a critical gap this study aims to address.

## Methods

### Data source and study design

This study is a retrospective cohort study using data from the MIMIC-IV database, a publicly available, de-identified electronic health records database that contains comprehensive clinical data from adult patients admitted to the ICU at Beth Israel Deaconess Medical Center in Boston, USA [15]. Author Guang Tu finished the CITI Data or Specimens Only Research course, obtained approval for database access, and assumed responsibility for data extraction (certification number 65828445). The study included adult ICU patients with a primary diagnosis of lymphoma. The diagnosis was based on the International Classification of Diseases, 10th Revision (ICD-10) codes. Exclusion criteria included missing key predictor or outcome variables" instead of the overly broad phrase, hospital stay less than 24 hours, and age under 18 years. The included patient data covered hospitalizations from 2008 to 2019 (Fig 1). The study utilized fully de-identified data from the publicly available MIMIC-IV database. Because the dataset lacks direct patient identifiers, the institutional

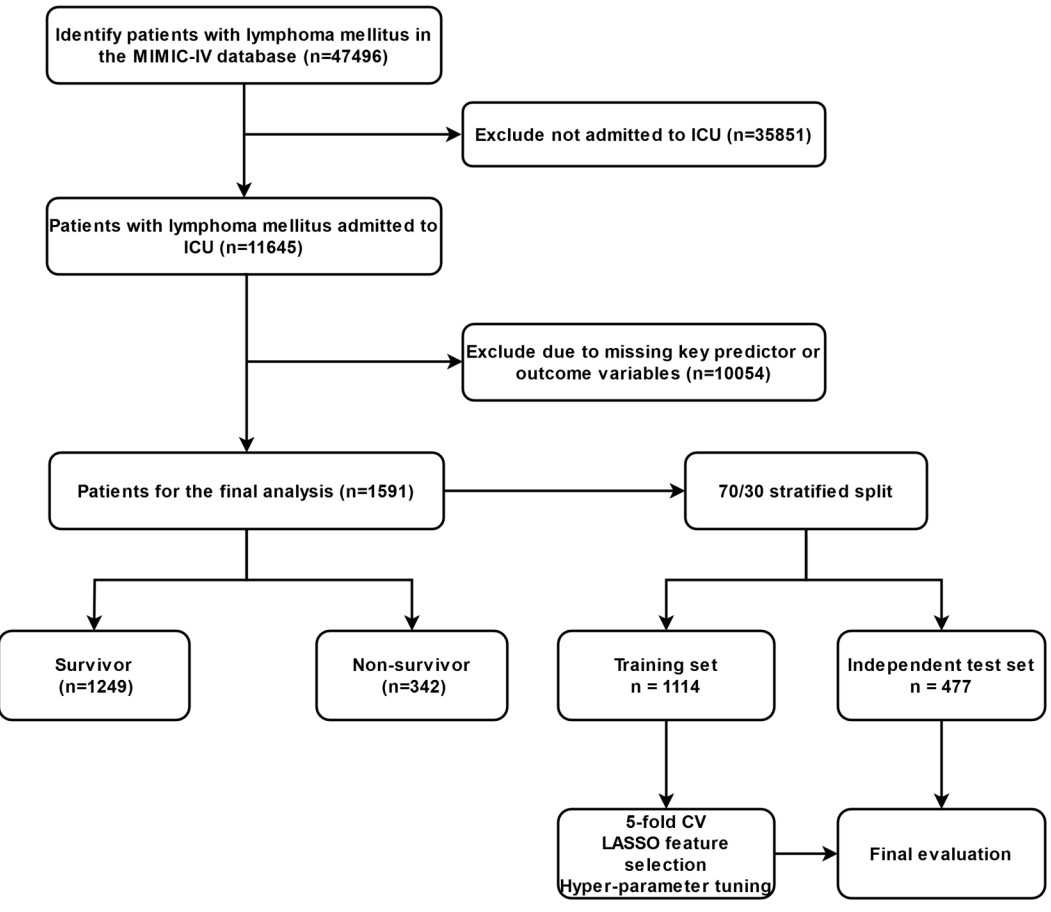

**Fig 1. Flowchart of patient inclusion.**

review boards of the authors' institutions determined that this research is exempt from IRB review and informed consent (Helsinki Declaration, 1964 and its later amendments).

## Data collection and processing

Baseline characteristics, laboratory parameters and clinical outcomes were extracted from MIMIC-IV. Prior to any modelling, the proportion of missing values for each variable was quantified (S1 Table). Across the 1 591 included patients, missingness ranged from 0% (gender, age, comorbidities, hospital-mortality flag) to 11.1% (APTT). Laboratory variables with the highest missing proportions were INR (9.9%), PT (9.9%), calcium (4.3%) and temperature (2.2%). The missing-data pattern was examined using Little's MCAR test ($\chi^2 = 2\ 184.7$, df = 1 959, $p = 0.002$), suggesting data were not missing completely at random. To minimise bias, we performed multivariate imputation by chained equations (MICE) with predictive mean matching for continuous variables and logistic/ polytomous regression for categorical variables, generating 20 imputed datasets (m = 20) under the missing-at-random (MAR) assumption. Each machine-learning model was then fitted independently on every imputed dataset; Rubin's rules were applied to pool performance metrics (AUC, accuracy, F1-score). Sensitivity analyses restricted to complete cases yielded similar AUCs (< 2% difference), indicating that the imputation procedure did not materially distort model discrimination.

## Outcome definition

In this study, in-hospital mortality is defined as death occurring at any time between ICU admission and hospital discharge, consistent with the standard definition adopted by the MIMIC-IV database and prior ICU outcome studies [15–17]. This endpoint was selected to capture the full spectrum of acute and subacute mortality risk during the hospital stay, aligning with clinical decision-making contexts where ICU prognostication directly influences treatment intensity and goals-of-care discussions.

## Risk factor selection

To identify risk factors significantly associated with in-hospital mortality, we used Lasso regression for variable selection. Lasso regression is a regularized linear regression method that introduces an L1 penalty term to automatically select variables with significant effects on the dependent variable. We determined the regularization parameter (λ) for Lasso regression through cross-validation and visualized the variable selection process using coefficient path and cross-validation plots. Ultimately, Lasso regression identified key risk factors including blood urea nitrogen, platelets, PT, heart rate, systolic blood pressure, APTT, spo2, and bicarbonate.

## Machine learning model development

Based on the key risk factors selected by Lasso regression, we developed 15 machine learning models, including logistic regression, random forest, gradient boosting, neural networks, and CatBoost Classifier. To ensure an unbiased estimate of model performance, the entire dataset was first subjected to a single stratified split into a 70% training set and a 30% hold-out test set based on the in-hospital mortality label. LASSO-based feature selection, and Bayesian hyper-parameter tuning (100 trials maximizing 5-fold cross-validated AUC) were performed exclusively within the training set using nested 5-fold cross-validation to prevent information leakage. The final CatBoost classifier, incorporating the selected features and optimized hyper-parameters, was then retrained on the full 70% training set and evaluated only once on the untouched 30% test set.

To address the 21.5% positive-class imbalance, we combined class weighting and resampling: models supporting class weights (CatBoost, Logistic, Ridge, SVM, LightGBM, XGBoost, Gradient Boosting) received minority-class weight w = N_majority/ N_minority ≈ 3.65 in their loss functions; the remaining models (Random Forest, Extra Trees, AdaBoost, MLP, Naive Bayes, KNN, Decision Tree) were trained after RandomOverSampler expanded the minority class to match the

majority within each cross-validation fold, preventing data leakage, while all reported metrics (AUC, accuracy, F1-score) were computed on the untouched, imbalanced test set.

## Model interpretation and validation

To enhance the clinical interpretability of the models, we used SHAP values for model interpretation. SHAP values, based on game theory, quantify the contribution of each variable to model predictions and reveal complex interactions among variables. Through SHAP value analysis, we not only validated the key variables identified by Lasso regression but also further evaluated their specific impacts on individual predictions. Moreover, SHAP waterfall plots decomposed the prediction results for individual patients, clearly showing the positive or negative contributions of each variable to the prediction outcomes, thereby providing a tool for clinicians to conduct individualized risk assessments.

## Statistical analysis

All statistical analyses were performed using R Statistical Software (Version 4.2.2) and the Free Statistics Analysis Platform (Version 2.1) [15]. A *P*-value less than 0.05 was considered statistically significant. Data processing and model training were completed on a local server to ensure data security and privacy.

## Results

### Baseline characteristics

A total of 1591 patients with lymphoma admitted to the ICU were included in this study, with 342 (21.5%) in-hospital deaths. The baseline characteristics of the patients are detailed in Table 1. Significant differences were observed between the survivor and non-survivor groups across multiple variables. The mean age of non-survivors (70.1 years) was slightly higher than that of survivors (69.1 years, *P* = 0.220), though not significantly. Non-survivors exhibited significantly higher heart rate, lower systolic and diastolic blood pressure, lower SpO2, lower hematocrit and hemoglobin levels, lower platelet counts, higher anion gap, lower bicarbonate levels, higher BUN, lower calcium levels, higher INR, longer PT, and longer APTT compared to survivors. The frequency and percentage of missing values for all variables are reported in S1 Table.

### Risk factor identification

Lasso regression was used to identify significant risk factors associated with in-hospital mortality. The screening process is illustrated in Fig 2, with the coefficient path plot (Panel A) and cross-validation plot (Panel B). The identified risk factors included blood urea nitrogen, platelets, PT, heart rate, systolic blood pressure, APTT, spo2, and bicarbonate. These factors were subsequently used as key input variables for the development of machine learning models.

**A: Coefficient path plot of Lasso regression; B: Cross-validation plot of Lasso regression.**

### Model performance

The performance of 15 machine learning models in predicting in-hospital mortality is summarized in Table 2. The CatBoost Classifier achieved the highest area under the receiver operating characteristic curve (AUC) of 0.7766, with an accuracy of 79.24% and an F1 score of 0.344. The Random Forest Classifier (AUC, 0.7691) and Extra Trees Classifier (AUC, 0.7667) also demonstrated strong performance. The ROC curves for all models are shown in Fig 3, intuitively reflecting their discriminatory power. The CatBoost Classifier exhibited superior performance in distinguishing between high-risk and low-risk patients.

### Variable importance and model interpretation

To evaluate potential multicollinearity among the eight variables retained by LASSO, we computed a Pearson correlation matrix (S1 Fig). The strongest absolute correlations were 0.60 between BUN and creatinine and 0.40 between platelet

**Table 1. Baseline characteristics of the patients.**

| Variables | Total (n = 1591) | Survivors (n = 1249) | Non-survivors (n = 342) | t/ Chi-square value | P _value |
|---|---|---|---|---|---|
| gender, n (%) | | | | 0.006 | 0.940 |
| female | 696 (43.7) | 547 (43.8) | 149 (43.6) | | |
| male | 895 (56.3) | 702 (56.2) | 193 (56.4) | | |
| age (year), mean (SD) | 69.3±13.6 | 69.1±13.6 | 70.1±13.3 | 1.505 | 0.220 |
| heart rate(beats/min), mean (SD) | 74.7±16.6 | 73.4±15.6 | 79.5±19.0 | 37.154 | < 0.001 |
| sbp (mmHg), mean (SD) | 89.8±17.6 | 91.5±16.6 | 83.5±19.6 | 57.315 | < 0.001 |
| dbp (mmHg), mean (SD) | 46.9±12.0 | 47.7±11.4 | 43.9±13.6 | 27.686 | < 0.001 |
| Temperature(°C), Mean±SD | 36.3±0.7 | 36.4±0.7 | 36.2±0.8 | 13.769 | < 0.001 |
| spo2, Mean±SD | 90.8±7.0 | 91.5±5.8 | 88.5±9.9 | 49.763 | < 0.001 |
| hematocrit(mg/dL), Mean±SD | 28.7±6.6 | 29.1±6.7 | 27.2±6.0 | 22.287 | < 0.001 |
| hemoglobin(mg/dL), mean (SD) | 9.4±2.2 | 9.6±2.3 | 8.8±2.0 | 31.559 | < 0.001 |
| platelets(×10⁹/L), mean (SD) | 148.0 (84.0, 231.0) | 158.0 (99.0, 239.0) | 101.0 (38.0, 188.0) | 58.955 | < 0.001 |
| wbc(×10⁹/L), Median (IQR) | 8.1 (4.9, 12.2) | 8.1 (5.0, 11.8) | 8.1 (4.1, 13.9) | 0.032 | 0.859 |
| aniongap(mg/dL), mean (SD) | 12.9±3.7 | 12.7±3.4 | 13.7±4.5 | 20.433 | < 0.001 |
| bicarbonate(mmol/dL), Mean±SD | 21.7±4.9 | 22.2±4.6 | 20.0±5.6 | 57.076 | < 0.001 |
| bun(mg/dL), Median (IQR) | 20.0 (13.0, 31.0) | 18.0 (12.0, 28.0) | 28.5 (19.0, 43.8) | 105.37 | < 0.001 |
| calcium(mmol/dL), mean (SD) | 8.2±0.9 | 8.3±0.9 | 8.0±1.0 | 27.475 | < 0.001 |
| chloride(mmol/dL), mean (SD) | 101.1±6.4 | 101.0±6.0 | 101.6±7.8 | 2.201 | 0.138 |
| creatinine(mg/dL), Median (IQR) | 0.9 (0.7, 1.3) | 0.9 (0.7, 1.3) | 1.1 (0.7, 1.6) | 35.995 | < 0.001 |
| glucose(mmol/dL), mean (SD) | 120.4±46.8 | 118.7±42.5 | 126.5±59.6 | 7.328 | 0.007 |
| sodium(mmol/dL), mean (SD) | 136.4±5.2 | 136.4±5.0 | 136.6±5.8 | 0.720 | 0.396 |
| potassium(mmol/dL), mean (SD) | 3.9±0.6 | 3.9±0.6 | 3.9±0.7 | 0.176 | 0.675 |
| inr, mean (SD) | 1.4±0.6 | 1.3±0.5 | 1.6±0.7 | 42.486 | < 0.001 |
| pt (s), mean (SD) | 15.0±5.8 | 14.4±5.1 | 16.9±7.5 | 46.924 | < 0.001 |
| aptt (s), mean (SD) | 32.8±14.8 | 31.8±13.7 | 36.4±18.0 | 23.519 | < 0.001 |
| myocardial_infarct, n (%) | | | | 2.503 | 0.114 |
| no | 1320 (83.0) | 1046 (83.7) | 274 (80.1) | | |
| yes | 271 (17.0) | 203 (16.3) | 68 (19.9) | | |
| heart_failure, n (%) | | | | 9.328 | 0.002 |
| no | 1050 (66.0) | 848 (67.9) | 202 (59.1) | | |
| yes | 541 (34.0) | 401 (32.1) | 140 (40.9) | | |
| peripheral_vascular, n (%) | | | | 2.698 | 0.100 |
| no | 1454 (91.4) | 1149 (92) | 305 (89.2) | | |
| yes | 137 (8.6) | 100 (8) | 37 (10.8) | | |
| dementia, n (%) | | | | 0.041 | 0.840 |
| no | 1547 (97.2) | 1215 (97.3) | 332 (97.1) | | |
| yes | 44 (2.8) | 34 (2.7) | 10 (2.9) | | |
| cerebrovascular, n (%) | | | | 1.454 | 0.228 |
| no | 1389 (87.3) | 1097 (87.8) | 292 (85.4) | | |
| yes | 202 (12.7) | 152 (12.2) | 50 (14.6) | | |
| chronic pulmonary disease, n (%) | | | | 0.061 | 0.805 |
| no | 1253 (78.8) | 982 (78.6) | 271 (79.2) | | |
| yes | 338 (21.2) | 267 (21.4) | 71 (20.8) | | |
| rheumatic disease, n (%) | | | | 0.001 | 0.970 |
| no | 1544 (97.0) | 1212 (97) | 332 (97.1) | | |

*(Continued)*

**Table 1.** (Continued)

| Variables | Total (n = 1591) | Survivors (n = 1249) | Non-survivors (n = 342) | t/ Chi-square value | P _value |
|---|---|---|---|---|---|
| yes | 47 (3.0) | 37 (3) | 10 (2.9) | | |
| peptic ulcer disease, n (%) | | | | 0.061 | 0.804 |
| no | 1546 (97.2) | 1213 (97.1) | 333 (97.4) | | |
| yes | 45 (2.8) | 36 (2.9) | 9 (2.6) | | |
| mild liver disease, n (%) | | | | 21.356 | < 0.001 |
| no | 1428 (89.8) | 1144 (91.6) | 284 (83) | | |
| yes | 163 (10.2) | 105 (8.4) | 58 (17) | | |
| diabetes, n (%) | | | | 4.098 | 0.043 |
| no | 1293 (81.3) | 1028 (82.3) | 265 (77.5) | | |
| yes | 298 (18.7) | 221 (17.7) | 77 (22.5) | | |
| paraplegia, n (%) | | | | 2.047 | 0.153 |
| no | 1506 (94.7) | 1177 (94.2) | 329 (96.2) | | |
| yes | 85 (5.3) | 72 (5.8) | 13 (3.8) | | |
| renal disease, n (%) | | | | 2.338 | 0.126 |
| no | 1257 (79.0) | 997 (79.8) | 260 (76) | | |
| yes | 334 (21.0) | 252 (20.2) | 82 (24) | | |
| malignant cancer, n (%) | | | | 9.899 | 0.002 |
| no | 293 (18.4) | 250 (20) | 43 (12.6) | | |
| yes | 1298 (81.6) | 999 (80) | 299 (87.4) | | |
| severe liver disease, n (%) | | | | 6.554 | 0.010 |
| no | 1527 (96.0) | 1207 (96.6) | 320 (93.6) | | |
| yes | 64 (4.0) | 42 (3.4) | 22 (6.4) | | |
| metastatic solid tumor, n (%) | | | | 0.082 | 0.775 |
| no | 1512 (95.0) | 1188 (95.1) | 324 (94.7) | | |
| yes | 79 (5.0) | 61 (4.9) | 18 (5.3) | | |
| aids, n (%) | | | | 0.684 | 0.408 |
| no | 1543 (97.0) | 1209 (96.8) | 334 (97.7) | | |
| yes | 48 (3.0) | 40 (3.2) | 8 (2.3) | | |

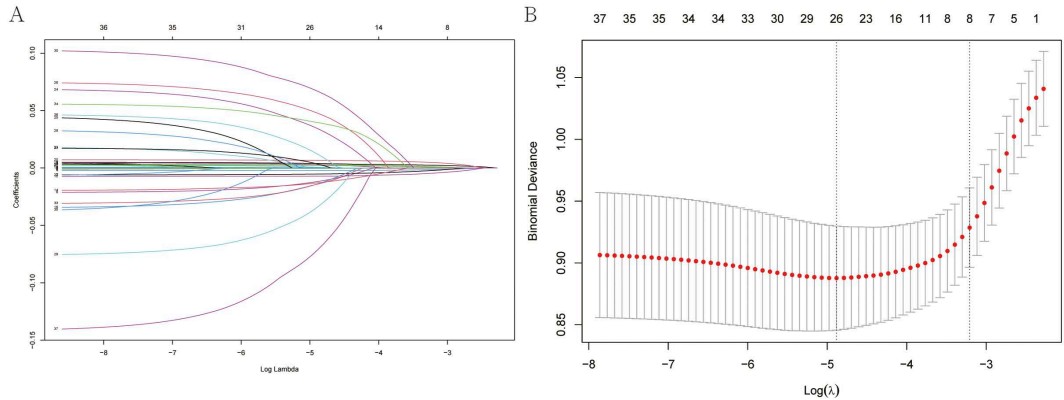

**Fig 2. Lasso regression screening results for in-hospital mortality risk factors in patients with lymphoma.**

**Table 2. Performance of each model for prediction.**

| Algorithm | AUC(%) | Accuracy(%) | F1score | predictive(%) |
|---|---|---|---|---|
| CatBoost Classifier | 0.7766 | 0.7924 | 0.344 | 0.5318 |
| Random Forest Classifier | 0.7691 | 0.8005 | 0.3451 | 0.5814 |
| Extra Trees Classifier | 0.7667 | 0.7987 | 0.307 | 0.5831 |
| Ridge Classifier | 0.765 | 0.7835 | 0.2087 | 0.5 |
| Linear Discriminant Analysis | 0.765 | 0.7862 | 0.28 | 0.5227 |
| Logistic Regression | 0.7622 | 0.7871 | 0.261 | 0.5252 |
| MLP Classifier | 0.7538 | 0.7808 | 0.2687 | 0.4893 |
| Naive Bayes | 0.7518 | 0.7727 | 0.3504 | 0.4617 |
| Gradient Boosting Classifier | 0.7482 | 0.7861 | 0.369 | 0.5025 |
| Light Gradient Boosting Machine | 0.7352 | 0.7871 | 0.367 | 0.5053 |
| SVM – Linear Kernel | 0.7157 | 0.7862 | 0.2489 | 0.5826 |
| Extreme Gradient Boosting | 0.7157 | 0.7718 | 0.3448 | 0.4426 |
| Ada Boost Classifier | 0.6999 | 0.7646 | 0.3389 | 0.4316 |
| K Neighbors Classifier | 0.6569 | 0.7655 | 0.262 | 0.4186 |
| Decision Tree Classifier | 0.6108 | 0.7251 | 0.3922 | 0.3769 |

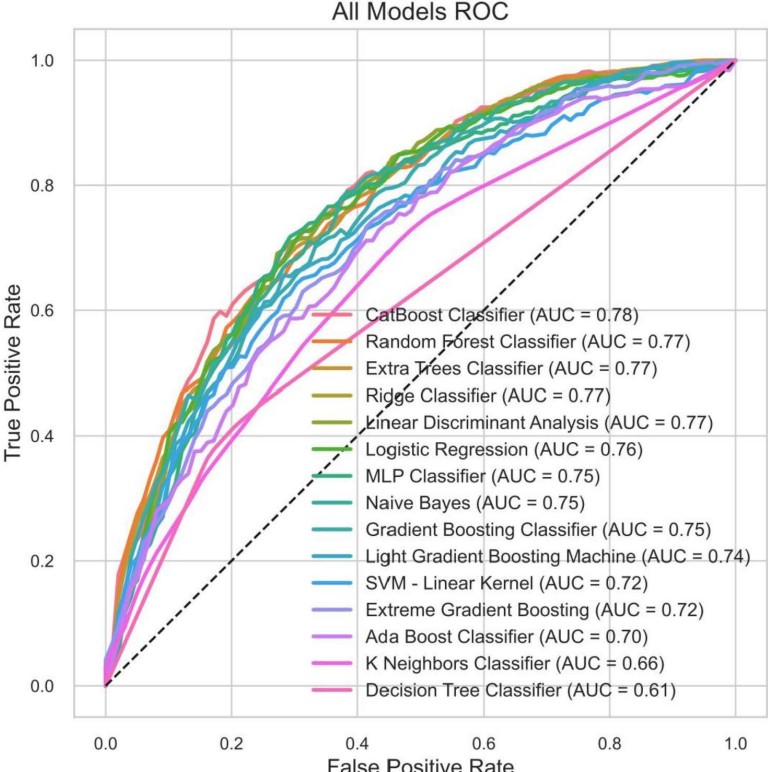

**Fig 3. Receiver Operating Characteristic curves of 15 models for in-hospital mortality in patients with lymphoma.**

count and PT; all remaining pairwise correlations were below 0.30. These low-to-moderate values indicate that multicollinearity is not a concern after LASSO regularisation, supporting the stability of the final model.

The importance weights of each variable in the models are displayed in Fig 4. Variables such as blood urea nitrogen, platelets, and PT were assigned higher weights, indicating their critical roles in predicting mortality. SHAP value analysis, shown in Fig 5, quantified the contribution of each variable to model predictions, validating the importance of key variables identified by Lasso regression. The SHAP waterfall plot (Fig 6) decomposed the prediction results for individual patients, clearly showing the positive or negative contributions of each variable to the prediction outcomes.

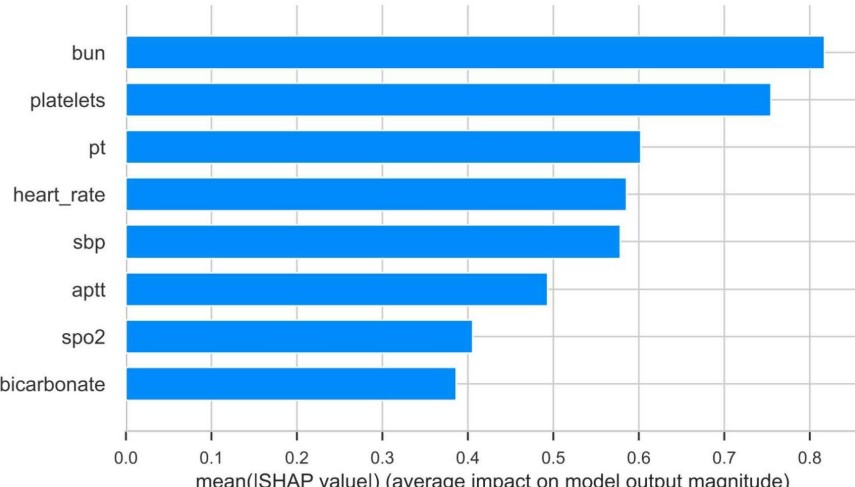

**Fig 4. The weights of variables importance.**

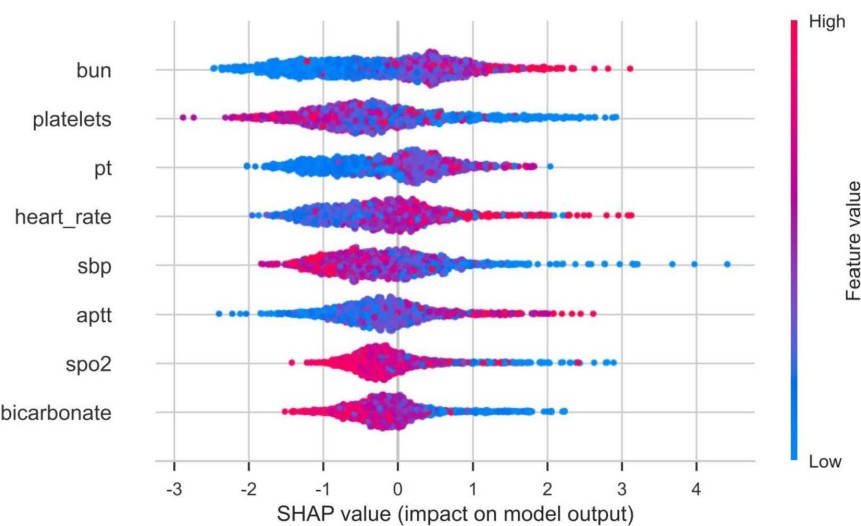

**Fig 5. The SHapley Additive exPlanation (SHAP) values.**

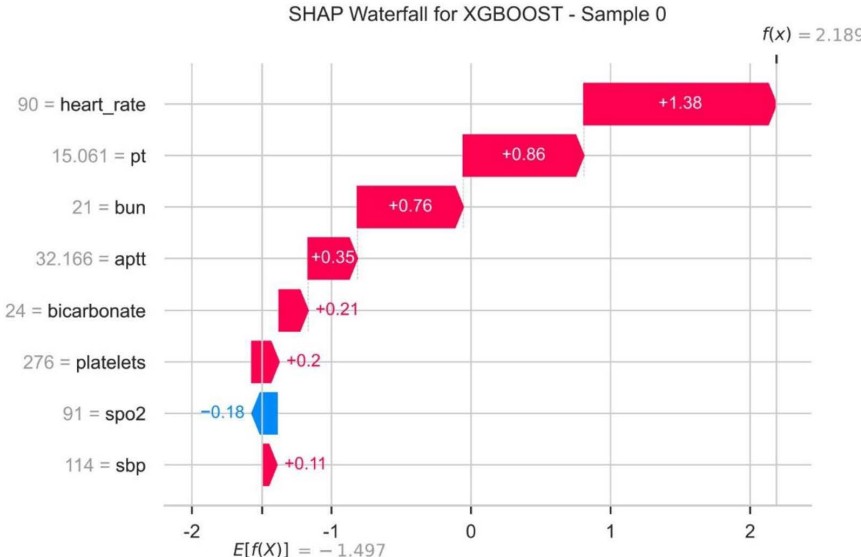

**Fig 6. The SHapley Additive exPlanations (SHAP) Waterfall.**

## Discussion

Our study successfully developed and validated machine learning (ML) models to predict in-hospital mortality among ICU patients with lymphoma. This achievement provides significant support for optimizing treatment strategies and improving patient outcomes. Traditional risk stratification tools are limited in their ability to handle the complex interactions among multiple clinical variables, whereas ML models, with their powerful data processing and pattern recognition capabilities, can provide more accurate predictions [9]. In this study, the CatBoost Classifier showed moderate discrimination, with an AUC of 0.7766, with potential incremental value over traditional scores and offering a new tool for clinical decision-making.

Compared with general medical or even oncologic ICU cohorts, lymphoma mellitus patients exhibit distinctive biological and therapeutic determinants of mortality risk. First, rapid tumour-lysis syndrome and hyper-metabolic states can precipitate acute kidney injury, directly elevating BUN independent of hypovolaemia or sepsis [16]. Second, marrow infiltration or ongoing cytotoxic chemotherapy frequently induces severe thrombocytopenia that is both more sudden and profound than in other malignancies [17]. Third, anthracycline-based regimens and immune checkpoint inhibitors increase the incidence of chemotherapy-associated cardiomyopathy, arrhythmias, and capillary leak, leading to haemodynamic instability (lower systolic blood pressure, higher heart rate) and coagulopathy (prolonged PT/APTT) [18,19]. Finally, opportunistic infections common in lymphoma (e.g., Pneumocystis jirovecii, invasive moulds) generate hypoxaemia (lower SpO$_2$) and lactic acidosis (lower bicarbonate) that may be disproportionate to the apparent severity of organ failure [20,21]. These lymphoma-specific factors collectively confound traditional scores such as APACHE or SOFA, which were calibrated on mixed ICU populations without accounting for tumour burden or chemotherapy-related toxicities.

Previous research has primarily focused on the application of machine learning in disease diagnosis and treatment optimization, with limited exploration into predicting in-hospital mortality for specific conditions like lymphoma in the ICU [22–25]. Our study fills this gap. Unlike previous studies that relied on a single model, we compared 15 different ML models and identified the CatBoost Classifier as the best performer. Moreover, through SHAP value analysis, we not only validated the key variables identified by Lasso regression but also revealed their specific impacts on individual predictions, providing clinicians with a more intuitive risk assessment tool. SHAP waterfall plots translate these individual explanations into immediate bedside actions: elevated BUN or severe thrombocytopenia prompts urgent nephrology

consultation for renal-replacement therapy ± rasburicase and haematology review for platelet transfusion or chemotherapy dose adjustment; a large positive SHAP contribution from low $SpO_2$ triggers intensified respiratory monitoring and early bronchoscopy with targeted anti-Pneumocystis or antifungal therapy; when cumulative SHAP values exceed a predefined threshold (e.g., > 0.6), the ICU team can initiate goals-of-care discussions—turning the waterfall plot into a real-time, pathophysiology-based action checklist. In this study, several variables significantly associated with in-hospital mortality were identified through Lasso regression and SHAP value analysis, including blood urea nitrogen, platelets, PT, heart rate, systolic blood pressure, APTT, $SpO_2$, and bicarbonate. These variables are not only statistically significant but also potentially related to the pathophysiological mechanisms of lymphoma. Elevated blood urea nitrogen levels may indicate renal dysfunction, In the lymphoma context, BUN elevation is often multifactorial: tumour-lysis–induced urate nephropathy [26], cisplatin or ifosfamide nephrotoxicity [27], and sepsis-related acute tubular necrosis all contribute [28]. Similarly, thrombocytopenia reflects not only bone marrow suppression but also immune-mediated platelet destruction (e.g., Evans syndrome) and chemotherapy-induced myelosuppression [29,30]. Prolonged PT and APTT mirror both disseminated intravascular coagulation triggered by tumour tissue factor expression and drug-induced coagulopathy (l-asparaginase, anthracyclines) [31]. These mechanistic links underscore why the CatBoost model, trained on lymphoma-specific data, outperforms generic scores that weight these variables identically across all ICU patients. A common complication in lymphoma patients that can be related to tumor burden, chemotherapy-induced nephrotoxicity, or infection [21,32,33]. Thrombocytopenia may suggest bone marrow suppression or disseminated intravascular coagulation (DIC), both of which are common in lymphoma patients and associated with disease severity and poor prognosis [34,35]. Additionally, changes in heart rate and blood pressure may reflect circulatory instability, while decreased $SpO_2$ may indicate respiratory failure, both of which are critical conditions frequently encountered in ICU patients [36–38]. By leveraging ML models, we can link these clinical variables to biological mechanisms, further elucidating the underlying causes of mortality risk in lymphoma patients admitted to the ICU. Future external validation using multicenter cohorts (e.g., eICU or regional ICU networks) is warranted to confirm the generalizability of our findings beyond the MIMIC-IV setting. Subsequent studies should incorporate longitudinal tumour-burden markers (LDH, PET-CT metabolic tumour volume) and detailed chemotherapy histories to refine lymphoma-specific mortality prediction beyond the current model.

Our study has several strengths. First, the data were sourced from the MIMIC-IV database, which contains a wealth of detailed clinical information, providing a solid foundation for model development and validation. Second, we employed multiple ML models and comprehensively evaluated their performance through cross-validation and SHAP value analysis, ensuring the reliability and interpretability of the results. Finally, our study not only focused on overall model performance but also assessed individual patient risks through SHAP force plots, offering more specific guidance for clinical applications.

## Limitations

Despite the positive outcomes, our study has some limitations. First, MIMIC-IV is a single-center database derived from Beth Israel Deaconess Medical Center (Boston, USA), which may introduce selection bias and limit generalizability to other ICUs with different patient demographics, clinical practices, or resource availability. Future research should conduct external validation on multiple independent datasets to confirm the generalizability of the models. Second, although ML models can identify key risk factors, their predictive mechanisms remain complex and difficult to fully explain using traditional medical theories. Additionally, our study did not consider the impact of treatments and interventions during the ICU stay on patient outcomes, which may limit the precision of the model's predictions. Because MIMIC-IV lacks lymphoma-specific variables such as histological subtype, Ann Arbor stage, recent chemotherapy exposure, tumour-lysis syndrome, or neutrophil nadir, these high-risk features could not be included in the analysis; future multicenter studies with dedicated oncology-ICU databases are needed to capture these predictors. Given the absence of granular treatment data (e.g., specific chemotherapy agents, immunotherapies, or evolving ICU bundles) in MIMIC-IV, we could not control

for temporal changes in lymphoma management or critical-care practices. Future multicenter studies that incorporate longitudinal drug and intervention data are warranted to address this limitation. Additionally, because MIMIC-IV lacks detailed, time-stamped data on ICU interventions (mechanical ventilation, vasopressors, dialysis, or lymphoma-specific treatments administered after admission), our models are restricted to admission/early-ICU variables and cannot perform dynamic risk reassessment. Future linkage with high-resolution treatment logs is required to develop longitudinal prediction frameworks.

## Conclusion

In summary, the ML models, particularly the CatBoost Classifier, may assist in risk estimation in-hospital mortality among ICU patients with lymphoma. These models not only offer additional insights alongside traditional approaches but also provide interpretable risk assessments through SHAP value analysis. Future research should focus on external validation and clinical implementation to improve outcomes for this high-risk patient population.

## Supporting information

**S1 Data. The complete, de-identified raw dataset for all patients who met the study inclusion and exclusion criteria.**
(XLS)

**S1 Table. Extent of missing data before imputation (n = 1591).**
(DOCX)

**S1 Fig. Correlation matrix of the final selected variables.**
(TIF)

## Author contributions

**Conceptualization:** Tianbi Lan, Ling Xu.

**Data curation:** Ling Xu, Guang Tu, Zhonglan Cai.

**Formal analysis:** Ling Xu, Guang Tu, Zhonglan Cai.

**Supervision:** Tianbi Lan.

**Writing – original draft:** Tianbi Lan, Ling Xu.

**Writing – review & editing:** Tianbi Lan.

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
