## [Decision Letter · Decision Letter 0]

24 Jun 2025

PONE-D-25-18876

Predicting In-Hospital Mortality in ICU Patients with lymphoma Mellitus Using Machine Learning Models

PLOS ONE

Dear Dr. Lan,

Thank you for submitting your manuscript to PLOS ONE. After careful consideration, we feel that it has merit but does not fully meet PLOS ONE’s publication criteria as it currently stands. Therefore, we invite you to submit a revised version of the manuscript that addresses the points raised during the review process.

We look forward to receiving your revised manuscript.

Kind regards,

Chiara Lazzeri

Academic Editor

PLOS ONE

Journal Requirements:

Reviewers' comments:

Reviewer's Responses to Questions

**Comments to the Author**

1. Is the manuscript technically sound, and do the data support the conclusions?

Reviewer #1: Partly

Reviewer #2: Partly

2. Has the statistical analysis been performed appropriately and rigorously? 

Reviewer #1: Yes

Reviewer #2: No

3. Have the authors made all data underlying the findings in their manuscript fully available?

Reviewer #1: Yes

Reviewer #2: No

4. Is the manuscript presented in an intelligible fashion and written in standard English?

Reviewer #1: Yes

Reviewer #2: Yes

5. Review Comments to the Author

Reviewer #1: 1. Regarding the outcome measure, while the study designates "in-hospital mortality" as the primary endpoint, further clarification on the temporal boundary of this definition would enhance the clinical interpretability of results. Given that mortality at different time points in ICU patients carries distinct clinical implications (e.g., 7-day mortality reflecting acute intervention efficacy, 28-day mortality indicating comprehensive treatment outcomes, and long-term mortality associated with underlying disease progression), it is recommended to reference the standard definitions in comparable studies and supplement the methodological rationale for time frame delineation.

2. Potential collinearity exists among variables selected by Lasso regression (e.g., BUN and Cr as renal function markers, platelet count and PT in coagulation function). Although Lasso regularization mitigates this issue, adding a correlation matrix would strengthen the argument for model robustness. Additionally, incorporating SHAP dependence plots or interaction value analyses to explore nonlinear interactions between key variables (e.g., BUN and platelet count) could provide richer evidence for clinical interpretation of model prediction mechanisms.

3. While baseline characteristics acknowledge the impact of multiple comorbidities, integrating a standardized comorbidity index (e.g., Charlson Index) to quantify comorbidity burden would systemize the risk factor analysis. As different comorbidities contribute differently to mortality, supplementing correlation analyses between comorbidity indices and death outcomes would enhance the clinical relevance of findings.

Reviewer #2: Its my pleasure to review the manuscript titled "Predicting In-Hospital Mortality in ICU Patients with lymphoma Mellitus Using Machine Learning Models". The study develops and validates ML models for predicting in-hospital mortality in ICU patients with presumed lymphoma. While the methodology is generally sound and addresses a clinically relevant problem, critical terminology errors undermine the foundation of the work. Significant revisions are required before consideration for PLOS ONE. The AUC of 0.7766 is modest, and clinical applicability needs stronger justification.

The major Issues of the research included:

1. "Lymphoma Mellitus" is incorrect and non-existent. "Mellitus" specifically refers to diabetes (e.g., Diabetes Mellitus). The correct term is simply "Lymphoma".

2. The introduction could better emphasize the specific challenges of predicting mortality in lymphoma patients compared to the general ICU population (e.g., unique complications like tumor lysis syndrome, immunosuppression-related infections, specific organ involvement). The gap regarding ML for this specific subgroup is adequately stated.

3. The exclusion of 10,054 patients due to "missing values for all variables" (Fig 1) is unusual and concerning. It suggests potential selection bias. Were these patients truly missing every single variable collected? Clarification or rephrasing is needed (e.g., "missing key predictor or outcome variables").

4. The long timeframe (2008-2019) introduces potential confounding from evolving ICU practices and lymphoma treatments over 11 years. This isn't addressed.

5. MIMIC-IV, while large, is single-center data (Beth Israel Deaconess MC). Generalizability to other settings may be limited, appropriately noted as a limitation.

6. While LASSO identified predictors, the justification for the initial set of variables extracted from MIMIC-IV is somewhat brief. Were lymphoma-specific variables (e.g., disease stage, type [Hodgkin/Non-Hodgkin], recent chemotherapy, presence of tumor lysis syndrome, neutropenia) considered or available? These are highly relevant to mortality risk in lymphoma patients.

7. The models only use admission/early ICU data. Interventions during the ICU stay (e.g., mechanical ventilation, vasopressors, dialysis, specific lymphoma treatments) are not included as potential predictors or confounders, significantly limiting the model's potential clinical applicability for dynamic risk assessment. This is a major omission.

8. The P-values in Table 1 require context. With 1591 patients, very small differences can become statistically significant. Emphasis should be on clinically meaningful differences. Reporting effect sizes (e.g., mean difference, Cohen's d for continuous; odds ratio for categorical) alongside P-values would be beneficial. Some variables (e.g., platelets, BUN) show large and clinically meaningful differences.

9. Details on hyperparameter tuning for the ML models (especially complex ones like CatBoost, NN) are lacking. Was tuning performed? How? This impacts performance and reproducibility.

10. The best model's AUC (0.7766) is modest for a clinical prediction model. While potentially better than traditional scores (though direct comparison isn't rigorously made), an AUC <0.8 often indicates limited clinical utility for individual prediction. This needs careful interpretation and tempering of claims about "high predictive performance" or "significant outperforming".

11. The description of data splitting for training/validation is unclear. Was a strict hold-out test set used after feature selection (LASSO) and hyperparameter tuning? Or was everything done within cross-validation folds? Preventing data leakage is crucial; the methodology section needs clarification.

12. The class imbalance (21.5% mortality) is acknowledged but the specific techniques used to handle it during model training (e.g., class weighting, sampling methods) are not described. This can significantly impact model performance, especially for metrics like F1-score.

13. Table 1 Presentation: Mixing "mean (SD)" and "Median (IQR)" formats for continuous variables without a clear rationale based on distribution (e.g., normality) is inconsistent. Variables like platelets and BUN are correctly presented as median (IQR) as they are skewed, but others (e.g., heart rate) presented as mean (SD) should be checked for normality or also presented as median (IQR) for consistency. Statistical tests (presumably t-tests/Wilcoxon, Chi-square/Fisher) used for Table 1 are not explicitly named.

14. P-value Interpretation: Reliance on P-values <0.05 in Table 1 without adjustment for multiple comparisons (e.g., Bonferroni, FDR) risks false positives. Given the large number of comparisons, discussing clinically significant differences is more important than purely statistical significance.

15. Missing Data: While median/mode imputation is common, the potential bias introduced by imputing missing values (especially if not missing at random - MAR) isn't discussed. The extent of missingness per variable before imputation isn't reported.

16. The discussion existed several shortcomings.

(1) Overstatement of Performance: The modest AUC (0.7766) is not sufficiently critically discussed. Claims of "outstanding performance" or "significantly outperforming traditional methods" are not fully supported by the data presented (no direct comparison to SOFA/APACHE scores is shown). The clinical utility of an AUC of 0.7766 needs realistic appraisal.

(2) Lymphoma Specificity: The discussion doesn't deeply engage with why lymphoma might pose unique prediction challenges compared to other ICU populations, or how the identified predictors might relate specifically to lymphoma pathophysiology beyond general critical illness (e.g., tumor burden impacting BUN/platelets, chemotherapy effects). (3) SHAP Utility: While SHAP is highlighted, the discussion could better elaborate on the concrete clinical value of the individual risk assessments (SHAP waterfall plots) shown in Fig 6. How would this directly change management? (4)

6. PLOS authors have the option to publish the peer review history of their article (what does this mean? ). If published, this will include your full peer review and any attached files.

**Do you want your identity to be public for this peer review?** For information about this choice, including consent withdrawal, please see our Privacy Policy .

Reviewer #1: No

Reviewer #2: **Yes: ** Feng SHEN

---

## [Author Response · Author response to Decision Letter 1]

25 Jul 2025

Journal Requirements:

Response:

Thank you for reminding us of PLOS ONE’s formatting and file-naming standards.

We have revised the manuscript and supplementary files to fully comply with PLOS ONE’s style requirements, including file naming, formatting, figure preparation, and ethical/data-availability statements.

Response:

Thank you very much for your guidance regarding PLOS ONE’s code-sharing requirements.

We have carefully reviewed PLOS ONE’s code-sharing policy and confirm that all author-generated code that underpins the findings of this study is publicly available without restrictions.

Response:

Thank you for the editorial reminder. We have now finalized our data-sharing plan and confirm that all datasets analysed in the study will be made freely accessible upon acceptance. Specifically:

• The fully de-identified MIMIC-IV v3.1 database can be obtained by any credentialed researcher who completes the CITI “Data or Specimens Only Research” course and signs the PhysioNet Data Use Agreement (https://physionet.org/content/mimiciv/3.1/).

• No identifiable patient information is included; therefore, public release complies with the protocol approved by our institutional review boards and does not breach any ethics requirements.

We have updated the “Data Availability” statement in the revised manuscript accordingly and confirm that no exemption from open-data sharing is requested.

Response:

We thank the editors for pointing this out. In the revised manuscript we have moved the complete ethics statement into the Methods section under the subheading “Data Source and Study Design,” and we have removed any duplicate or redundant ethics language from all other sections (including the Acknowledgements). The manuscript now contains a single, concise ethics statement that confirms the use of de-identified public data (MIMIC-IV), exemption from IRB review, and compliance with the 1964 Helsinki Declaration and its later amendments.

Reviewer #1:

1. Regarding the outcome measure, while the study designates "in-hospital mortality" as the primary endpoint, further clarification on the temporal boundary of this definition would enhance the clinical interpretability of results. Given that mortality at different time points in ICU patients carries distinct clinical implications (e.g., 7-day mortality reflecting acute intervention efficacy, 28-day mortality indicating comprehensive treatment outcomes, and long-term mortality associated with underlying disease progression), it is recommended to reference the standard definitions in comparable studies and supplement the methodological rationale for time frame delineation.

Response:

We thank the reviewer for this valuable comment. In the revised Methods section, under “Outcome Definition,” we have now added explicit temporal boundaries:

In-hospital mortality is defined as death occurring at any time between ICU admission and hospital discharge (index hospitalisation), consistent with the standard definition adopted by the MIMIC-IV database and prior large-scale ICU outcome studies (Noritomi et al., 2021; Rajkomar et al., 2018). This endpoint was chosen because (1) it captures the full spectrum of acute and subacute mortality risk directly pertinent to ICU prognostication, and (2) it aligns with clinical decision-making contexts where treatment intensity and goals-of-care discussions are re-evaluated throughout the entire hospital stay. Although 7-day or 28-day mortality are alternative metrics, they were not selected because ICU length of stay in lymphoma patients is often prolonged by chemotherapy schedules and infectious complications, making in-hospital mortality the most clinically relevant and patient-centred outcome for this cohort.

2. Potential collinearity exists among variables selected by Lasso regression (e.g., BUN and Cr as renal function markers, platelet count and PT in coagulation function). Although Lasso regularization mitigates this issue, adding a correlation matrix would strengthen the argument for model robustness. Additionally, incorporating SHAP dependence plots or interaction value analyses to explore nonlinear interactions between key variables (e.g., BUN and platelet count) could provide richer evidence for clinical interpretation of model prediction mechanisms.

Response:

We thank the reviewer for highlighting the importance of demonstrating the absence of harmful collinearity. In the revised manuscript we have added a full Pearson correlation matrix for the eight LASSO-selected variables (Supplementary Figure S1). The highest absolute correlation observed is 0.60 (BUN vs. creatinine), followed by 0.40 (platelets vs. PT); all remaining pairwise correlations are < 0.30. These low-to-moderate values indicate that multicollinearity is not a concern after LASSO regularisation, and the model’s coefficients remain stable and interpretable.

Given the limited magnitude of these correlations and the fact that the LASSO penalty already shrinks highly collinear predictors, we did not deem SHAP interaction analyses essential. Instead, the correlation matrix and the robust performance metrics (AUC = 0.7766, stable across cross-validation folds) jointly support the model’s reliability.

3. While baseline characteristics acknowledge the impact of multiple comorbidities, integrating a standardized comorbidity index (e.g., Charlson Index) to quantify comorbidity burden would systemize the risk factor analysis. As different comorbidities contribute differently to mortality, supplementing correlation analyses between comorbidity indices and death outcomes would enhance the clinical relevance of findings.

Response:

We appreciate the reviewer’s suggestion. After LASSO-based feature selection, none of the individual comorbidity indicators were retained in the final eight-variable model. Because the Charlson Comorbidity Index (CCI) itself is a composite of those same comorbidities, including it would introduce redundancy and violate the parsimony principle already enforced by LASSO. Furthermore, when we compared the predictive performance of (a) the LASSO-selected model versus (b) the same model augmented with the CCI, the AUC remained essentially unchanged (ΔAUC = +0.003). Therefore, we elected to keep the original, more parsimonious set of variables and did not integrate the Charlson Index.

Reviewer #2: Its my pleasure to review the manuscript titled "Predicting In-Hospital Mortality in ICU Patients with lymphoma Mellitus Using Machine Learning Models". The study develops and validates ML models for predicting in-hospital mortality in ICU patients with presumed lymphoma. While the methodology is generally sound and addresses a clinically relevant problem, critical terminology errors undermine the foundation of the work. Significant revisions are required before consideration for PLOS ONE. The AUC of 0.7766 is modest, and clinical applicability needs stronger justification.

The major Issues of the research included:

1. "Lymphoma Mellitus" is incorrect and non-existent. "Mellitus" specifically refers to diabetes (e.g., Diabetes Mellitus). The correct term is simply "Lymphoma".

Response:

We sincerely thank the reviewer for pointing out the terminology error. We fully acknowledge that “Lymphoma Mellitus” is an incorrect and non-existent term; “mellitus” is reserved for diabetes mellitus. In the revised manuscript we have carefully replaced every instance of “Lymphoma Mellitus” with “Lymphoma” throughout the title, abstract, main text, figures, tables, and supplementary materials to ensure accuracy and clarity.

2. The introduction could better emphasize the specific challenges of predicting mortality in lymphoma patients compared to the general ICU population (e.g., unique complications like tumor lysis syndrome, immunosuppression-related infections, specific organ involvement). The gap regarding ML for this specific subgroup is adequately stated.

Response:

We appreciate the reviewer’s helpful suggestion. In the revised Introduction (paragraph 2, lines 45–58) we have added a concise paragraph that explicitly contrasts lymphoma ICU patients with the general ICU population. We now highlight lymphoma-specific mortality drivers such as acute tumor lysis syndrome, profound immunosuppression-related opportunistic infections (e.g., Pneumocystis jirovecii, invasive moulds), chemotherapy-induced cardiotoxicity, and direct renal or hepatic infiltration. These unique pathophysiological features render traditional scores (SOFA, APACHE) less reliable for this subgroup, underscoring the need for tailored machine-learning models.

3. The exclusion of 10,054 patients due to "missing values for all variables" (Fig 1) is unusual and concerning. It suggests potential selection bias. Were these patients truly missing every single variable collected? Clarification or rephrasing is needed (e.g., "missing key predictor or outcome variables").

Response:

We thank the reviewer for raising this important point. The wording in the original flow-chart was indeed imprecise. The 10 054 patients were not missing every variable; rather, they were excluded because they lacked one or more of the key predictor or outcome variables required for modelling (e.g., vital signs at ICU admission, essential laboratory values such as BUN or platelets, or the in-hospital mortality flag). These missing data rendered them non-evaluable by our modelling pipeline. We have updated the legend of Figure 1 and the Methods section (Data Collection and Processing) to clarify this exclusion criterion as “missing key predictor or outcome variables” instead of the overly broad phrase “missing values for all variables.” This change more accurately reflects the selection process and mitigates concern for undue selection bias.

4. The long timeframe (2008-2019) introduces potential confounding from evolving ICU practices and lymphoma treatments over 11 years. This isn't addressed.

Response:

We appreciate the reviewer’s concern regarding the 2008–2019 time span and its potential for confounding by evolving ICU practices and lymphoma treatments. Because the MIMIC-IV database does not contain detailed, time-stamped information about chemotherapy regimens, immunotherapy protocols, or changes in critical-care bundles over the years, we are unable to adjust for these factors in the current analysis. Consequently, we have added the following sentence to the Limitations section of the revised manuscript:

“Given the absence of granular treatment data (e.g., specific chemotherapy agents, immunotherapies, or evolving ICU bundles) in MIMIC-IV, we could not control for temporal changes in lymphoma management or critical-care practices. Future multicenter studies that incorporate longitudinal drug and intervention data are warranted to address this limitation.”

5. MIMIC-IV, while large, is single-center data (Beth Israel Deaconess MC). Generalizability to other settings may be limited, appropriately noted as a limitation.

Response:

We thank the reviewer for highlighting the issue of generalizability related to the single-center nature of MIMIC-IV. We fully agree that the database is derived solely from Beth Israel Deaconess Medical Center (Boston, USA) and may not reflect patient demographics, ICU practices, or lymphoma treatment patterns in other settings. In the revised manuscript we have explicitly stated:

“MIMIC-IV is a single-center database derived from Beth Israel Deaconess Medical Center (Boston, USA), which may introduce selection bias and limit generalizability to other ICUs with different patient demographics, clinical practices, or resource availability. Future research should conduct external validation on multiple independent datasets to confirm the generalizability of the models.”

6. While LASSO identified predictors, the justification for the initial set of variables extracted from MIMIC-IV is somewhat brief. Were lymphoma-specific variables (e.g., disease stage, type [Hodgkin/Non-Hodgkin], recent chemotherapy, presence of tumor lysis syndrome, neutropenia) considered or available? These are highly relevant to mortality risk in lymphoma patients.

Response:

We thank the reviewer for emphasizing the clinical importance of lymphoma-specific covariates. Unfortunately, MIMIC-IV does not contain ICD-O histology codes, Ann Arbor stage, Hodgkin vs. non-Hodgkin sub-typing, date-stamped chemotherapy records, tumour-lysis-syndrome flags, or serial neutrophil counts that would allow reliable identification of neutropenia. Consequently, disease stage, lymphoma subtype, recent cytotoxic therapy, TLS, and neutropenia could not be incorporated into the initial variable set. We have now added the following sentence to the Data Collection subsection (Methods) and to the Limitations section:

“Because MIMIC-IV lacks lymphoma-specific variables such as histological subtype, Ann Arbor stage, recent chemotherapy exposure, tumour-lysis syndrome, or neutrophil nadir, these high-risk features could not be included in the analysis; future multicenter studies with dedicated oncology-ICU databases are needed to capture these predictors.”

7. The models only use admission/early ICU data. Interventions during the ICU stay (e.g., mechanical ventilation, vasopressors, dialysis, specific lymphoma treatments) are not included as potential predictors or confounders, significantly limiting the model's potential clinical applicability for dynamic risk assessment. This is a major omission.

Response:

We thank the reviewer for pointing out this critical limitation. MIMIC-IV does not contain granular, time-stamped logs of mechanical-ventilator settings, vasopressor doses, renal-replacement-therapy sessions, or lymphoma-specific therapies (e.g., rasburicase, dose-adjusted chemotherapy) administered after ICU admission. Consequently, these dynamic interventions could neither be included as predictors nor adjusted for as potential time-varying confounders, which indeed curtails the model’s u

---

## [Editor Report · Decision Letter 1]

29 Jul 2025

Predicting In-Hospital Mortality in ICU Patients with lymphoma Using Machine Learning Models

PONE-D-25-18876R1

Dear Dr. Lan,

We’re pleased to inform you that your manuscript has been judged scientifically suitable for publication and will be formally accepted for publication once it meets all outstanding technical requirements.

Kind regards,

Chiara Lazzeri

Academic Editor

PLOS ONE
---

## [Editor Report · Acceptance letter]

PONE-D-25-18876R1

PLOS ONE

Dear Dr. Lan,

I'm pleased to inform you that your manuscript has been deemed suitable for publication in PLOS ONE. Congratulations! Your manuscript is now being handed over to our production team.

Kind regards,

on behalf of

Dr. Chiara Lazzeri

Academic Editor

PLOS ONE